# Prediction of Bone Metastasis in Prostate Cancer Using Blood Glucose-6-Phosphate Dehydrogenase Activity: A Retrospective Medical Record Review

**DOI:** 10.3390/cancers18010041

**Published:** 2025-12-23

**Authors:** Chang-Hwan Yeom, Jiewon Lee, Keun-Joo Bae, Kangseok Kim, Jongsoon Choi, Myeong-Hun Lim

**Affiliations:** 1Department of Family Medicine, YCH Hospital, Seoul 05700, Republic of Korea; lymphych@hanmail.net (C.-H.Y.); babble2@naver.com (J.L.); jjoo611@nate.com (K.-J.B.); 2Department of Surgery, Bangre Hospital, Incheon 22228, Republic of Korea; kangseok78@gmail.com; 3Department of Family Medicine, Kosin University College of Medicine, Busan 49267, Republic of Korea; fmcjs@naver.com

**Keywords:** prostate cancer, bone metastasis, G6PD activity, biomarker

## Abstract

Prostate cancer is the most common diagnosed cancer among men in the United States. Bone is the predominant metastatic site in prostate cancer, and bone metastases occur in more than 80% of patients with castration-resistant prostate cancer. Accurate and early prediction of bone metastasis in prostate cancer is crucial for deciding treatment decisions. This study aimed to evaluate the potential of G6PD activity level as a biomarker for predicting bone metastasis in patients with prostate cancer. We revealed fair diagnostic performance of G6PD activity and calculated optimal cutoff value of G6PD activity level. The results support value of G6PD activity as a non-invasive marker for metastatic risk assessment. We suggest that serial monitoring of G6PD activity may help assess the risk of bone metastasis and enable early detection in patients with prostate cancer.

## 1. Introduction

In the United States, prostate cancer is the most common diagnosed type of cancer among men with approximately 300,000 new cases reported in 2024 [1]. When prostate cancer metastasized to the bone, patients may experience various symptoms, including bone pain, along with skeletal-related events (SREs) such as pathological fractures and hypercalcemia [2,3]. In particular, pathologic fractures have been shown to negatively impact survival in patients with prostate cancer [4,5]. Also, early initiation of bone-targets therapy can help prevent SREs [6,7]. Thus, accurate prediction of bone metastasis in prostate cancer is essential for guiding treatment decisions, such as aggressive pain management, radiotherapy, and systemic therapies, and is also important for the prevention and management of related complications [8]. 

Currently, bone metastasis in prostate cancer is primarily diagnosed using imaging techniques such as bone scans and MRI, usually after symptoms have appeared, making early prediction of bone metastasis challenging [9,10]. Therefore, there is a need to utilize accessible blood-based biomarkers to monitor the presence of bone metastasis and enable early detection. 

Glucose-6-phosphate dehydrogenase (G6PD) is an essential enzyme for the production of NADPH, which is essential for protecting cells against oxidative stress by enabling the regeneration of reduced glutathione [11]. Recent studies have reported that cancer cells overexpress G6PD to increase NADPH level, thereby adapting to excess oxidative stress and promoting the synthesis of fatty acids required for rapid cell proliferation [12]. In prostate cancer, G6PD expression has been shown to increase with tumor progression, and metastatic prostate cancer exhibits higher level of G6PD expression compared to localized prostate cancer or benign prostatic tissue [13]. Furthermore, it has been reported that prostate cancer cells with bone metastasis exhibit increased G6PD expression in response to cytokine secreted by bone marrow stromal cells residing in the bone microenvironment [13]. Given its role in tumor progression and its regulation within the bone microenvironment, G6PD is a potential biomarker for predicting bone metastasis in prostate cancer. 

Therefore, this study aimed to evaluate the potential of G6PD activity level as a biomarker for predicting bone metastasis in patients with prostate cancer. We assessed the association between G6PD activity and the presence of bone metastasis and identified an optimal cutoff value to calculate sensitivity and specificity.

## 2. Methods

### 2.1. Patients

This study is a retrospective medical record review conducted at a single institution. A total of 56 patients participated in this study using consecutive sampling. This study included all prostate cancer patients aged 19 years or older who visited YCH Hospital between October 2018 and June 2025. Exclusion criteria were as follows: patients who did not undergo G6PD activity testing, patients with unknown bone metastasis status, and patients diagnosed with two or more different types of cancer. Through a retrospective review of medical records, we collected data on patients’ age, diagnosis of prostate cancer, bone metastasis status, and G6PD activity level. In addition, bone metastasis was defined based on the documented diagnosis in the patient’s medical records, including diagnostic certificates provided from external hospitals. The age and bone metastasis status of participants were assessed on the data the G6PD activity test was performed. 

### 2.2. G6PD Activity Test

G6PD activity was measured using the Dr. Rappeler G6PD & Hemoglobin system. Dr. Rappeler R1 G6PD & Hemoglobin systems (Dr. Rappeler, Anyang, Republic of Korea), which are approved by the Ministry of Food and Drug Safety as in vitro diagnostic medical devices, are designed for the quantitative measurement of total G6PD and specific G6PD enzyme activity in whole blood and capillary blood. In this study, capillary blood was used consistently for all measurements to ensure data uniformity. The measurements are performed using an electrochemical method on the Dr. Rappeler R1 Analyzer, which is CE-certified under the European IVDR. The system can have a test result in 4 minutes and is useful for point-of-care testing of G6PD activity. 

The quality control protocols of the system have undergone validation by regulatory agencies. The device shows validated quality control parameters and regulator-approved performance in the Instructions for Use. Reproducibility and repeatability testing showed coefficients of variation ranging from 3.92–16.61% and 3.81–18.05%, respectively. Accuracy and correlation analyses showed a correlation coefficient of 0.99, indicating a strong agreement with a comparative product. The device’s analytical sensitivity was rigorously verified, demonstrating a measurement range of 0–300 U/dL. The limit of detection (LOD) and limit of blank were 2 U/dL and 0 U/dL, respectively, and a limit of quantitation was 17 U/dL. 

The G6PD strip operates on the electrochemical principle by measuring the current generated through the reduction of NADP⁺ to NADPH, which is coupled with ferricyanide as an electron mediator. Based on the magnitude of the generated current, the total G6PD activity is calculated and expressed as units per deciliter (U/dL) of whole blood. This value is converted into a specific G6PD value (U/g Hb) by dividing the total G6PD activity (U/dL) by the hemoglobin concentration (g/dL).

### 2.3. Statistical Analysis

R version 4.4.2 was used for statistical analysis. First, the Mann–Whitney U test was used to compare G6PD activity levels according to presence of bone metastasis because of the skewed distribution of G6PD activity. For the comparison of age according to bone metastasis status, an independent t-test was used. Secondly, to identify independent predictors of bone metastasis, univariable and multivariable logistic regression analysis were performed using age and G6PD activity level as independent variables. The outcome variable was the presence or absence of bone metastasis. To evaluate the predictive performance of G6PD activity level for bone metastasis, a receiver operating characteristic (ROC) curve analysis was conducted. ROC curve was generated with the x-axis representing 1-specificity and the y-axis representing sensitivity for various cutoff values [14]. The area under the curve (AUC) and 95% confidence interval (95% CI) were calculated to assess discriminative ability. Finally, the optimal cutoff value was determined using Youden’s index. Youden’s index, calculated as Sensitivity + Specificity – 1, is used to determine the optimal cutoff value at the point where the index reached its maximum [15]. Corresponding sensitivity, specificity, positive predictive value (PPV), and negative predictive value (NPV) were presented at the optimal cutoff value. The statistical significance was defined as *p* < 0.05. 

## 3. Results

### 3.1. General Characteristics

A total of 56 patients participated in this retrospective medical record review study. Patients were categorized into two groups depending on the presence or absence of bone metastasis, and comparisons were made for age and G6PD activity level. The mean age was 66.6 years in the group without bone metastasis and 68.4 years in the group with bone metastasis, showing no significant difference between the two groups. There was a significant difference in median G6PD activity levels between the two groups (Table 1). Figure 1 showed the distribution of G6PD activity levels between patients with and without bone metastasis. The group with bone metastasis showed a median G6PD activity level of 12.2 U/g Hb, whereas the group without bone metastasis had a median level of 10.0 U/g Hb (Figure 1).

### 3.2. Logistic Regression Analysis for Predictors of Bone Metastasis

Univariable logistic regression analysis revealed that G6PD activity was a significant predictor of bone metastasis, whereas age was not significantly associated with bone metastasis. In the multivariable logistic regression analysis, G6PD activity remained the only independent predictor of bone metastasis, with an odds ratio (OR) of 1.08 (95% CI, 1.03–1.12). 

### 3.3. Receiver Operating Characteristic Curve for G6PD Activity

To evaluate the potential of G6PD activity level as a biomarker for predicting bone metastasis, a ROC curve was generated (Figure 2). The area under the curve (AUC) was 0.78 (95% CI, 0.66–0.90). Using Youden’s index, the optimal cutoff value of G6PD activity level for predicting bone metastasis was determined to be 11.5 U/g Hb, which maximized the value of sensitivity + specificity – 1 [15,16]. 

### 3.4. Diagnostic Performance of G6PD Activity

Table 2 presents the sensitivity, specificity, PPV, NPV for five different G6PD activity cutoff values. The cutoff of 11.5 U/g Hb demonstrated the best predictive performance, with a sensitivity of 0.81, specificity of 0.73, PPV of 0.54, and NPV of 0.91.

## 4. Discussion

In this study, we explored the clinical relevance of G6PD activity in predicting bone metastasis among prostate cancer patients. G6PD activity was significantly higher in patients with bone metastasis and remined an independent predictor in multivariable logistic regression analysis. In addition, ROC curve analysis further demonstrated fair diagnostic performance, supporting the potential of G6PD activity as a supplementary non-invasive marker for metastatic risk assessment [17]. These findings are consistent with previous report of elevated serum G6PD activity observed in patients with skin cancer. Notably, serum G6PD activity has been reported to increase with tumor stage and decrease after treatment in Merkel cell carcinoma, a type of skin cancer, which supports the broader clinical relevance of G6PD as a biomarker in oncology [18]. 

G6PD activity was significantly associated with bone metastasis in prostate cancer, with an OR of 1.08. Although the incremental increase per 1 U/g Hb is relatively small, the result demonstrates a consistent biological trend. Since G6PD activity varies across a wide range, a difference of 10 U/g Hb doubles the level of risk. Also, considering the non-invasive and convenient nature of G6PD activity test, the findings show potential to be combined with other biomarkers to improve predictive power of bone metastasis. 

Several other biomarkers and clinical parameters have been investigated for their potential to predict bone metastasis in prostate cancer [19]. Bone-specific alkaline phosphatase (BALP) and pro-collagen type I N-terminal propeptide (PINP) are well-established bone formation markers that have been extensively studied in the context of predicting bone metastasis [9]. BALP has demonstrated a sensitivity and specificity of 0.75 and 0.93, respectively, while PINP has shown values of 0.78 and 0.68 [20,21]. Both markers exhibited lower sensitivity compared to serum G6PD activity observed in the present study. On the other hand, prostate-specific antigen (PSA), a widely used biomarker in prostate cancer management, has demonstrated a sensitivity of 0.85 and a specificity of 0.84, both of which are higher than those of G6PD activity. However, its diagnostic accuracy can be compromised by benign condition such as benign prostatic hyperplasia, limiting its positive predictive value (PPV) in detecting bone metastasis [16]. In addition, Aihara et al. reported that PSA levels decrease in prostate cancers with a high Gleason score, which limits its ability to predict the progression of the disease [22]. G6PD activity has the advantage of predicting bone metastasis of prostate cancer through a simple blood test. However, a limitation remains in that G6PD levels may also increase in the presence of other undetected malignancies not related to prostate cancer. Furthermore, the positive predictive value is low, which limits its utility as independent screening tool for predicting bone metastasis. 

G6PD is a molecule with a molecular weight of 59 kDa and consists of 515 amino acids [23]. It plays multiple roles in cellular functions through the pentose phosphate pathway. G6PD plays a key role in cellular redox homeostasis by catalyzing the conversion of NADP⁺ to NADPH [24]. NADPH is essential for reducing oxidized glutathione, which in turn is necessary for reducing reactive oxygen species (ROS), including hydrogen peroxide, within the cell [25]. Therefore, G6PD, the rate-limiting enzyme in the NADPH-generating pathway, plays a critical role in protecting cells from oxidative stress. In addition, G6PD contributes to nucleotide biosynthesis by generating ribose-5-phosphate, a key precursor for DNA and RNA, through the pentose phosphate pathway [26].

Numerous studies have reported the role of G6PD in cancer progression, metastasis and proliferation in cancer cells [26]. Firstly, inhibition of G6PD in prostate cancer cells leads to decrease NADPH and RNA synthesis, along with increased reactive oxygen species (ROS) level [27]. Thus, G6PD is essential for rapidly dividing cancer cells, contributing to both nucleotide production and the maintenance of redox homeostasis [26]. Secondly, G6PD regulates genes such as LDHA, MT2A that promote proliferation and invasion and induces cellular migration in bone metastatic prostate cancer [13]. Thirdly, G6PD increases endothelial nitric oxide synthase (eNOS) activity through vascular endothelial growth factor (VEGF), promoting endothelial cell proliferation and migration, thereby playing a crucial role in angiogenesis [28]. 

Indeed, the expression and activity of G6PD increase with prostate cancer progression and are particularly elevated in bone metastatic prostate cancer. In animal models, G6PD expression increased with prostate cancer progression and remained elevated in castration-resistant prostate cancer [27]. G6PD activity in prostate cancer tissues positively correlates with the Gleason grade, which is commonly associated with prostate cancer aggressiveness and progression [29]. G6PD expression increases with advancing disease stage, and is particularly elevated in bone metastatic prostate cancer cells compared to benign prostate tissue or non-bone metastatic prostate cancer cells [13]. 

G6PD expression in prostate cancer is tightly regulated by both intrinsic oncogenic signaling and extrinsic cues from the tumor microenvironment. Firstly, androgen receptor (AR) signaling is fundamentally involved in both the onset and advancement of prostate cancer, as most prostate cancers rely on the activation of the AR pathway to sustain cellular survival and proliferation [30]. Recent study suggests that AR signaling promotes prostate cancer cell growth by increasing G6PD expression and enhancing flux through pentose phosphate pathway via activation of the mTOR-SREBP1 axis [27]. In addition, prostate cancer cells exhibit upregulation of G6PD within the bone microenvironment. Within the bone microenvironment, interleukin-6 (IL-6) secreted by bone marrow stromal cells has been shown to induce the upregulation of G6PD expression in prostate cancer cell [13]. In the setting of bone metastasis, G6PD expression in prostate cancer cell is upregulated by cytokine secreted by cells within the bone and through androgen receptor (AR) signaling. 

This study has several strengths. First, it is the first study to propose G6PD activity as a potential biomarker for predicting bone metastasis in prostate cancer. Numerous studies biologically demonstrated the association between G6PD activity and bone metastasis of prostate cancer [13,31]. These findings support the potential of G6PD as a biomarker in this context. In this study, we conducted ROC curve analysis to determine the cutoff value and calculate the performance metrics, including sensitivity and specificity, thereby demonstrating the potential clinical utility of G6PD activity in a quantitative manner. Second, the analytical method for measuring G6PD activity was standardized to improve the reproducibility of the results. Although various devices are available for measuring G6PD activity, this study exclusively utilized the R1 analyzer, which contributed to enhancing the reliability and consistency of the test results. Furthermore, the use of a finger prick sampling technique and rapid test time increased the accessibility and feasibility of the test in clinical settings. Unlike traditional G6PD activity test that rely on large equipment and are less feasible for follow-up, our method requires only a single drop of blood sample and a portable analyzer, allowing for easier short-term monitoring. 

Despite its strengths, this study has several limitations. First, we were unable to consider multiple covariates that may influence G6PD activity due to insufficient data. According to previous studies, infection, diabetes mellitus, hemolytic anemia, TNM stages and chemotherapy status could vary G6PD activity [32,33,34]. The lack of consideration of covariates that could affect G6PD levels has limited the internal validity of this study. Multivariable logistic regression should be performed to adjust for these covariates in future studies. Also, further investigation involving subgroup analysis stratified by PSA levels and Gleason score which is key prognostic markers for prostate cancer is needed. Second, the G6PD activity measured in this study was obtained from red blood cells (RBC). Nevertheless, there are biological limitations in explaining the relationship between cancer progression and G6PD activity measured in RBC, as opposed to G6PD activity within the cancer cells. Further research is needed to clarify the biological relationship between serum G6PD activity and tumor cell metabolism. Third, because the number of participants included in this study was very small and the study was conducted at a single center, the external validity is limited and generalizability of the findings is restricted. In addition, due to the cross-sectional design of our study, it is not possible to clearly evaluate changes in G6PD activity before and after the development of bone metastasis. Further analysis in a large cohort of patients is needed to analyzed temporal changes in G6PD and the relationship with disease progression. Although the sample size was relatively small, a post hoc power analysis yielded a statistical power of 0.88 based on the observed effect size. This indicates that the study maintained sufficient power to detect significant differences, supporting the statistical reliability of our findings. Furthermore, the positive predictive value (PPV) is low at 0.54, suggesting a high incidence of false positive. Even considering that the prevalence of bone metastasis in the overall prostate cancer group is not high, the use of G6PD as a standalone predictor for bone metastasis is limited. Thus, G6PD activity could be utilized as a supplementary biomarker or in combination with other markers. 

In the present study, we established a cutoff value of 11.5 U/g Hb for G6PD activity and calculated the corresponding sensitivity and specificity to evaluate its clinical applicability in predicting bone metastasis. These findings suggest that serial monitoring of G6PD activity may help assess the risk of bone metastasis and enable early detection in patients with prostate cancer. Furthermore, additional studies are needed to determine whether G6PD activity, when combined with other non-invasive biomarkers such as PSA, ALP and osteopontin, can improve the accuracy of predicting bone metastasis in prostate cancer patients. 

## 5. Conclusions

This study revealed clinical utility of G6PD activity as a biomarker for predicting bone metastasis in prostate cancer. We confirmed that G6PD activity was higher in patients with bone metastasis than patients without bone metastasis. Furthermore, this study calculated AUC and decided cutoff value of G6PD activity for predicting bone metastasis. This finding is consistent with findings from previous studies on skin cancer and provides supporting evidence for the use of the G6PD activity test as a biomarker for detecting bone metastasis in patients with prostate cancer. Further large-scale studies involving more participants are needed in prostate cancer, as well as additional research to evaluate its clinical applicability in other types of cancer. 

## Figures and Tables

**Figure 1 cancers-18-00041-f001:**
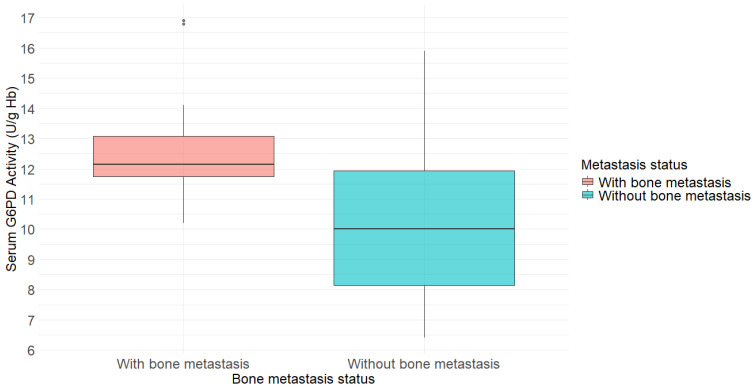
Boxplot comparing G6PD activity levels between prostate cancer patients with and without bone metastasis. The median G6PD activity was higher in the group with bone metastasis (12.2 U/g Hb) than in the group without bone metastasis (10.0 U/g Hb).

**Figure 2 cancers-18-00041-f002:**
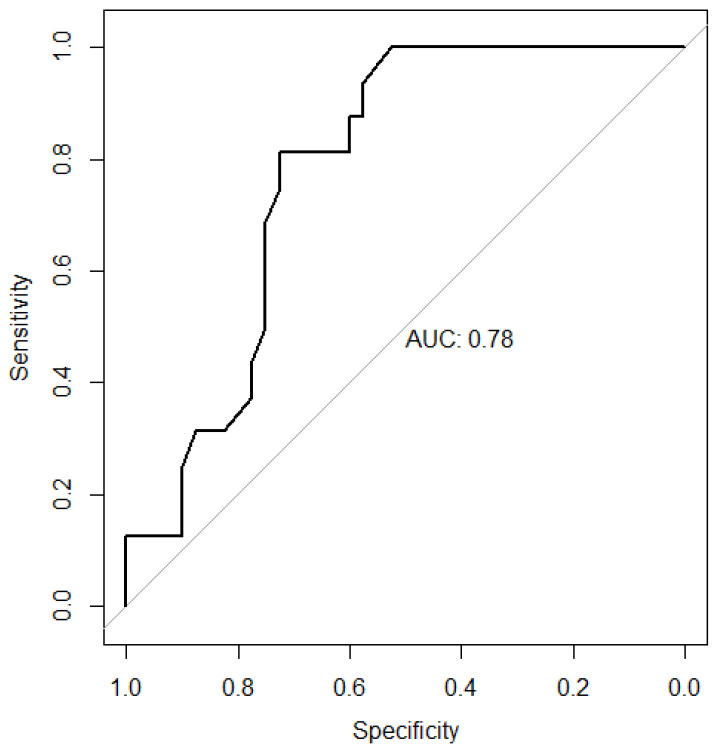
Receiver operating characteristic (ROC) curve for G6PD activity level in predicting bone metastasis in prostate cancer patients. The area under the curve (AUC) was 0.78 (95% CI, 0.66–0.90).

**Table 1 cancers-18-00041-t001:** General characteristics of participants according to bone metastasis.

	Without Bone Metastasis (N = 40)	With Bone Metastasis (N = 16)	*p*-Value
**Demographics**			
Age (years) ^1^	66.6 ± 8.4	68.4 ± 6.3	0.383
**Clinical outcomes**			
G6PD activity (U/g Hb) ^2^	10.0 (8.2–11.9)	12.2 (11.8–13.1)	0.001

^1^ Value was presented as mean ± standard deviation for age and the *p*-value was obtained by independent *t*-test. ^2^ Value was presented as median (interquartile range) for G6PD activity and the *p*-value was obtained by Mann–Whitney U test.

**Table 2 cancers-18-00041-t002:** Diagnostic performance of G6PD activity level for predicting bone metastasis at various cutoff values. Sensitivity, specificity, positive predictive value (PPV), and negative predictive value (NPV) are presented for each cutoff values.

Cutoffs for G6PD Activity (U/g Hb)	Sensitivity	Specificity	PPV	NPV
9.5	1.00	0.45	0.42	1.00
10.5	0.94	0.58	0.47	0.96
11.5	0.81	0.73	0.54	0.91
12.5	0.38	0.78	0.40	0.76
13.5	0.19	0.90	0.43	0.73

## Data Availability

The datasets presented in this article are not publicly available due to ethical and legal restrictions, as they contain pseudonymized patient information. Requests to access the data may be considered on a case-by-case basis and should be directed to lymphych@hanmail.net.

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
