# Peer review of "Prediction of Bone Metastasis in Prostate Cancer Using Blood Glucose-6-Phosphate Dehydrogenase Activity: A Retrospective Medical Record Review"

_cancers, 2025, doi:10.3390/cancers18010041_

Round 1

Reviewer 1 Report

Comments and Suggestions for Authors

Comments

The study is described as cross-sectional, yet the wording suggests a retrospective chart review. The authors must clearly define the study design and justify the appropriateness of using a cross-sectional methodology for evaluating a predictive biomarker.

The sample size (n = 56) is relatively small. No justification, power calculation, or effect-size consideration is provided. This raises concerns about statistical power and the generalizability of findings.

Details regarding how patients were selected from the medical records are insufficient. The authors should clarify whether consecutive sampling or convenience sampling was used to avoid selection bias.

Exclusion criteria are mentioned, but comorbidities and treatment history which may influence G6PD levels were not controlled or accounted for. This omission limits the internal validity of the study.

The manuscript does not sufficiently describe how bone metastasis was confirmed (bone scan, MRI, CT, PET?).

It should be indicated whether a standardized diagnostic criterion was applied and whether assessments were performed by blinded radiologists.

Although the device used (Dr. Rappeler R1 Analyzer) is described, there is insufficient information about: Calibration protocol, Quality control procedures, Inter-assay and intra-assay variability

The authors should specify whether capillary or venous blood was used consistently, as G6PD values can differ between sample types.

The manuscript presents only age and G6PD levels. Key clinical features (PSA levels, Gleason score, TNM stage, treatments) are missing. These variables are essential for comparing patient groups and assessing bias.

The OR of 1.08 per unit increase of G6PD is statistically significant but clinically small. The clinical relevance of such a small incremental increase should be discussed.

AUC of 0.78 indicates fair accuracy, not strong. This should be acknowledged.

PPV is low (0.54), indicating that false positives are common. This limits the clinical usefulness of G6PD as a stand-alone predictor.

The claim that G6PD is an “effective biomarker” is overstated given the small sample size, limited covariate adjustment, and moderate AUC.

The Discussion does not adequately address confounding factors that could alter G6PD levels independent of metastasis.

Below is the list of good work in this field, please cross discuss

doi: 10.56434/j.arch.esp.urol.20247701.10, doi: 10.56434/j.arch.esp.urol.20247703.32

https://doi.org/10.1016/j.ecoenv.2025.118209, https://doi.org/10.1111/wvn.70048, doi: 10.2174/1574893616666210708143556

Comments on the Quality of English Language

  • The overall English quality of the manuscript is acceptable, but several sections require significant editing for grammar, clarity, and scientific expression.

  • Numerous sentences are overly long or structurally complex, which affects readability. Shortening sentences and improving flow would enhance clarity.

  • There are multiple instances of incorrect verb tenses, subject–verb disagreement, and inconsistent use of articles (a/an/the). Careful grammatical revision is required throughout the manuscript.

Author Response

Reviewer #1:

We deeply thank you for taking the time to read our manuscript and to provide feedback. Your comments were highly insightful and helped us to improve the quality of our study. We addressed the concerns point by point in the following pages.

 Point P1

The study is described as cross-sectional, yet the wording suggests a retrospective chart review. The authors must clearly define the study design and justify the appropriateness of using a cross-sectional methodology for evaluating a predictive biomarker.

(Response)

We are really grateful for reviewer’s insightful comment. We agree that the terminology used in the original manuscript may have been confusing. We have revised the ‘cross-sectional design’ to ‘retrospective medical record review’ through out the manuscript to accurately reflect our data collection process. Also, regarding the use of a cross-sectional approach for evaluating G6PD as a predictive biomarker, we believe it is appropriate for the following reasons. The blood samples for G6PD activity measurement were collected at a time point closely aligned with the clinal diagnosis of bone metastasis. This allowed us to assess the immediate association between the biomarker level and the presence of metastasis. We have also explicitly addressed the limitations of our study in the discussion section. Specifically, we noted that the cross-sectional nature of the study limits our ability to determine temporal causality and acknowledged the relatively lower strength of evidence. We have emphasized the need for future large-scale prospective cohort studies to confirm our findings. We are thankful for your constructive comments that helped improve the quality of our study.

Line 282-288: “Third, because the number of participants included in this study was very small and the study was conducted at a single center, the external validity is limited and generalizability of the findings is restricted. In addition, due to the cross-sectional design of our study, it is not possible to clearly evaluate changes in G6PD activity before and after the development of bone metastasis. Further analysis in a large cohort of patients is needed to analyzed temporal changes in G6PD and the relationship with disease progression.”

 Point P2

The sample size (n = 56) is relatively small. No justification, power calculation, or effect-size consideration is provided. This raises concerns about statistical power and the generalizability of findings.

(Response)

We sincerely appreciate the reviewer’s valuable comments. We acknowledge the reviewer’s concern regarding the relatively small sample size. We have stated that the relatively small sample size may limit the generalizability of our results to a broader population. We have emphasized our study should be viewed as an exploratory analysis. In addition, we performed a post-hoc power analysis to verify the statistical validity of our findings. Based on the observed data, the calculated statistical power was 0.88 after adjusting for the efficiency of the Wilcoxen rank-sum test. The statistical power exceeds the standard threshold of 0.80. We added the result of statistical power in the discussion section.

Line 282-288: “Third, because the number of participants included in this study was very small and the study was conducted at a single center, the external validity is limited and generalizability of the findings is restricted. In addition, due to the cross-sectional design of our study, it is not possible to clearly evaluate changes in G6PD activity before and after the development of bone metastasis. Further analysis in a large cohort of patients is needed to analyzed temporal changes in G6PD and the relationship with disease progression.”

 Line 288-291: “Although the sample size was relatively small, a post-hoc power analysis yielded a statis-tical power of 0.88 based on the observed effect size. This indicates that the study maintained sufficient power to detect significant differences, supporting the statistical reliability of our findings.”

Point P3

Details regarding how patients were selected from the medical records are insufficient. The authors should clarify whether consecutive sampling or convenience sampling was used to avoid selection bias.

(Response)

We sincerely apologize for the reviewer’s comment. We also agree that the details regarding the selection method were insufficient. We have clarified the sampling method in the revised manuscript. We used consecutive sampling to avoid selection bias, including all patients who met the inclusion criteria during the study period. We sincerely appreciate reviewer’s valuable feedback regarding the sampling method.

Line 81-86: “This study is a retrospective medical record review conducted at a single institution. A total of 56 patients participated in this study using consecutive sampling. This study included all prostate cancer patients aged 19 years or older who visited YCH Hospital between October, 2018 and June, 2025. Exclusion criteria were as follows: patients who did not undergo G6PD activity testing, patients with unknown bone metastasis status, and patients diagnosed with two or more different types of cancer.”

Editor Point P4

Exclusion criteria are mentioned, but comorbidities and treatment history which may influence G6PD levels were not controlled or accounted for. This omission limits the internal validity of the study.

(Response)

We really appreciate for the reviewer’s advice. We agree with the reviewer’s comment that the failure to account of confounders, such as comorbidities and treatment history, is a limitation of this study. Due to the nature of retrospective chart review, comprehensive data on all possible variables were not consistently available. We agree that future studies incorporating a more robust control of these confounders are needed. We have added these points to limitation section. We are sincerely grateful for reviewer’s comment.

Line 270-276: “First, we were unable to consider multiple covariates that may influence G6PD activity due to insufficient data. According to previous studies, infection, diabetes mellitus, hemolytic anemia, TNM stages and chemotherapy status could vary G6PD activity [32-34]. The lack of consideration of covariates that could affect G6PD levels has limited the internal validity of this study. Multivariable logistic regression should be performed to adjust for these covariates in future studies.”

Editor Point P5

The manuscript does not sufficiently describe how bone metastasis was confirmed (bone scan, MRI, CT, PET?). It should be indicated whether a standardized diagnostic criterion was applied and whether assessments were performed by blinded radiologists.

(Response)

We are grateful for reviewer’s comment regarding operational definition of bone metastasis. We agree with the reviewer’s comment that a more specific operational definition of bone metastasis is necessary. In this study, bone metastasis was confirmed by thoroughly reviewing patient’s medical certificates, referral letters, and clinical records to determine whether a diagnosis of bone metastasis from prostate cancer was explicitly documented. We have updated the manuscript to clarify this definition. We are thankful for reviewer’s valuable guidance on enhancing the specificity of our operational definition.

Line 86-90: “Through a retrospective review of medical records, we collected data on patients’ age, diagnosis of prostate cancer, bone metastasis status, and G6PD activity level. In addition, bone metastasis was defined based on the documented diagnosis in the patient’s medical records, including diagnostic certificates provided from external hospitals.”

 Editor Point P6

Although the device used (Dr. Rappeler R1 Analyzer) is described, there is insufficient information about: Calibration protocol, Quality control procedures, Inter-assay and intra-assay variability

(Response)

We are grateful for reviewer’s comment regarding the device we used in this study. We agree with the reviewer’s comment that a more detailed description of the device is necessary. Accordingly, we have provided comprehensive information regarding the protocol and quality control procedures in the revised manuscript. We are thankful for reviewer’s valuable suggestion regarding the device description.

Line 101-110: “The quality control protocols of the system have undergone validation by regulatory agencies. The device shows validated quality control parameters and regulator-approved performance in the Instructions for Use. Reproducibility and repeatability testing showed coefficients of variation ranging from 3.92-16.61% and 3.81-18.05%, respectively. Accuracy and correlation analyses showed a correlation coefficient of 0.99, indicating a strong agreement with a comparative product. The device’s analytical sensitivity was rigorously verified, demonstrating a measurement range of 0-300 U/dL. The limit of detection (LOD) and limit of blank were 2 U/dL and 0 U/dL, respectively and a limit of quantitation was 17 U/dL.”

 Editor Point P7

The authors should specify whether capillary or venous blood was used consistently, as G6PD values can differ between sample types.

(Response)

We are grateful for reviewer’s comment regarding the blood sampling method. We completely agree with the reviewer’s comment regarding the impact of sample types of G6PD values. To maintain consistency, we used capillary blood for all participants. We have added this in the revised method section.

Line 97-98: “In this study, capillary blood was used consistently for all measurements to ensure data uniformity.”

 Editor Point P8

The manuscript presents only age and G6PD levels. Key clinical features (PSA levels, Gleason score, TNM stage, treatments) are missing. These variables are essential for comparing patient groups and assessing bias.

(Response)

We are sincerely thankful for reviewer’s comment. We fully agree with the reviewer’s comment that incorporating key clinical variables related to prostate cancer would significantly enhance the internal validity of the study. However, due to the retrospective nature of this study, we encountered practical limitation in data collection and were unable to include all relevant clinical parameters. We regret that these variables were not fully accounted for in the current analysis.

As suggested, we acknowledge that performing stratified analyses based on PSA levels, Gleason score, TNM stage and adjusting for treatment would further minimize potential bias. We have added a discussion on these limitations in the revised manuscript and are committed to conduct follow-up studies that incorporate these variables to ensure more robust results. We are thankful for pointing out crucial aspect of the research.

Line 270-277: “First, we were unable to consider multiple covariates that may influence G6PD activity due to insufficient data. According to previous studies, infection, diabetes mellitus, hemolytic anemia, TNM stages and chemotherapy status could vary G6PD activity [32-34]. The lack of consideration of covariates that could affect G6PD levels has limited the internal validity of this study. Multivariable logistic regression should be performed to adjust for these covariates in future studies. Also, further investigation involving subgroup analysis stratified by PSA levels and Gleason score which is key prognostic markers for prostate cancer is needed.”

 Editor Point P9

The OR of 1.08 per unit increase of G6PD is statistically significant but clinically small. The clinical relevance of such a small incremental increase should be discussed.

(Response)

We are really grateful for reviewer’s comment. We have addressed the information by clarifying that while per-unit increase is modest, the cumulative effect is clinically significant. For example, 10 U/g Hb increase in G6PD activity which is within the observed range of the sample, corresponds to a more than 2-folds increase in the risk of bone metastasis. Also, G6PD test is a non-invasive POCT method and the results shows value as a supplementary tool for predicting bone metastasis when integrated with existing clinical markers.

Line 190-195: “G6PD activity was significantly associated with bone metastasis in prostate cancer, with an OR of 1.08. Although the incremental increase per 1 U/g Hb is relatively small, the result demonstrates a consistent biological trend. Since G6PD activity varies across a wide range, a difference of 10 U/g Hb doubles the level of risk. Also, considering the non-invasive and convenient nature of G6PD activity test, the findings show potential to be combined with other biomarkers to improve predictive power of bone metastasis.”

 Editor Point P10

AUC of 0.78 indicates fair accuracy, not strong. This should be acknowledged.

(Response)

We sincerely appreciate the reviewer’s constructive comment regarding the interpretation of the AUC value. We agree that an AUC value of 0.78 corresponds to a fair diagnostic performance, and it is important to describe its clinical significance accurately. Accordingly, we have revised the manuscript to acknowledge this level of performance. We have also added a discussion highlighting that the results show possibility as a supplementary biomarker that can provide added value when used in conjunction with existing clinical parameters.

Line 183-185: “In addition, ROC curve analysis further demonstrated fair diagnostic performance, sup-porting the potential of G6PD activity as a supplementary non-invasive marker for meta-static risk assessment [17].”

 Editor Point P11

PPV is low (0.54), indicating that false positives are common. This limits the clinical usefulness of G6PD as a stand-alone predictor.

(Response)

We are grateful for reviewer’s comment regarding positive predictive value. We entirely agree that the low PPV may result in a high false positive rate. Consequently, we acknowledge the limitations of using G6PD activity as a standalone predictor. In response to reviewer’s comment, we have updated the manuscript to show that G6PD could be used as a supplementary biomarker or in combination with other biomarkers. We sincerely appreciate reviewer pointing out this critical issue.

Line 291-295: “Furthermore, the positive predictive value (PPV) is low at 0.54, suggesting a high incidence of false positive. Even considering that the prevalence of bone metastasis in the overall prostate cancer group is not high, the use of G6PD as a standalone predictor for bone metastasis is limited. Thus, G6PD activity could be utilized as a supplementary biomarker or in combination with other markers.”

 Editor Point P12

The claim that G6PD is an “effective biomarker” is overstated given the small sample size, limited covariate adjustment, and moderate AUC.

(Response)

We are thankful for reviewer’s suggestion. Considering the limitations of the study and the AUC analysis, we agree that labeling G6PD activity as an effective biomarker may be an overstatement. Therefore, we have revised its classification to a complementary biomarker. We are grate for reviewer’s valuable comment.

Line 41-43: “G6PD activity is a complementary non-invasive biomarker for predicting bone metastasis in patients with prostate cancer.”

 Editor Point P13

The Discussion does not adequately address confounding factors that could alter G6PD levels independent of metastasis.

(Response)

We are grateful for reviewer’s comment regarding factors that affect G6PD. We apologize for not fully accounting for the various factors that could influence G6PD levels, which is crucial for a highly reliable analysis. Due to initial data constraints, several potential confounders were not sufficiently addressed. We have conducted a comprehensive literature review and updated the limitation section to include a detailed discussion of factors affecting G6PD activity.

Line 270-276: “First, we were unable to consider multiple covariates that may influence G6PD activity due to insufficient data. According to previous studies, infection, diabetes mellitus, hemolytic anemia, TNM stages and chemotherapy status could vary G6PD activity [30-32]. The lack of consideration of covariates that could affect G6PD levels has limited the internal validity of this study. Multivariable logistic regression should be performed to adjust for these covariates in future studies.”

 Editor Point P14

Below is the list of good work in this field, please cross discuss

doi: 10.56434/j.arch.esp.urol.20247701.10,

doi: 10.56434/j.arch.esp.urol.20247703.32

https://doi.org/10.1016/j.ecoenv.2025.118209,

https://doi.org/10.1111/wvn.70048

doi: 10.2174/1574893616666210708143556

(Response)

We appreciate reviewer’s comment regarding great research about prostate cancer. We have updated the introduction section to include a wider range of previous studies regarding prostate cancer. The additions provide a more comprehensive overview of the various research.

Line 196-197: “Several other biomarkers and clinical parameters have been investigated for their potential to predict bone metastasis in prostate cancer [18].”

 Line 51-56: “In particular, pathologic fractures have been shown to negatively impact survival in patients with prostate cancer [4,5]. Also, early initiation of bone-targets therapy can help prevent SREs [6,7]. Thus, accurate prediction of bone metastasis in prostate cancer is essential for guiding treatment decisions, such as aggressive pain management, radiotherapy, and systemic therapies, and is also important for the prevention and management of related complications [8].”

Reviewer 2 Report

Comments and Suggestions for Authors

The article addresses a clinically important issue of early identification of bone metastases in prostate cancer patients and introduces a novel biomarker—G6PD activity—as a potential screening tool. Among the study’s strengths are the clearly presented biological rationale for choosing G6PD (its role in oxidative stress, proliferation, progression, and specific activation in the bone microenvironment) and the coherent results: higher median G6PD activity in patients with metastases, statistical significance in logistic regression, and a moderately good AUC value (0.78), allowing the determination of a cutoff point at 11.5 U/g Hb. The authors used a standardized G6PD assay and presented a clear ROC analysis with sensitivity, specificity, and predictive values, which enhances transparency and potential clinical applicability. Another strength is the well-grounded discussion of previously used biomarkers (BALP, PINP, PSA) and the contextual placement of G6PD among them.

The study’s weaknesses stem mainly from design limitations. The sample size is very small (N = 56) and single-center, which limits the generalizability of the findings. The cross-sectional design precludes assessing prognostic value over time. The authors lacked key confounding variables that could affect G6PD activity—such as infections, diabetes, hemolytic anemia, and chemotherapy—which they acknowledge; the lack of control for these factors limits the credibility of the conclusions. Another limitation is that G6PD was measured in erythrocytes rather than serum or tumor tissue; as the authors correctly note, this makes the biological interpretation challenging since it is unclear whether RBC activity reflects tumor metabolism. Additionally, several authors have commercial ties to the production of the G6PD test, and a related patent has been filed—this represents a significant conflict of interest that could influence data interpretation. Finally, the PPV values are low, which limits the clinical utility of the test as a standalone screening tool.

In summary, the study presents an interesting and promising biomarker with solid biological rationale and consistent data, but its significance is constrained by small sample size, lack of clinical confounder control, the biomarker’s measurement method, and potential conflicts of interest. The results should be considered preliminary and require confirmation in larger, prospective studies.

Comments on the Quality of English Language

English is fine.

Author Response

Reviewer #2:

We deeply thank you for taking the time to read our manuscript and to provide feedback. Your comments were highly insightful and helped us to improve the quality of our study. We addressed the concerns point by point in the following pages.

 Point P1

The article addresses a clinically important issue of early identification of bone metastases in prostate cancer patients and introduces a novel biomarker—G6PD activity—as a potential screening tool. Among the study’s strengths are the clearly presented biological rationale for choosing G6PD (its role in oxidative stress, proliferation, progression, and specific activation in the bone microenvironment) and the coherent results: higher median G6PD activity in patients with metastases, statistical significance in logistic regression, and a moderately good AUC value (0.78), allowing the determination of a cutoff point at 11.5 U/g Hb. The authors used a standardized G6PD assay and presented a clear ROC analysis with sensitivity, specificity, and predictive values, which enhances transparency and potential clinical applicability. Another strength is the well-grounded discussion of previously used biomarkers (BALP, PINP, PSA) and the contextual placement of G6PD among them.

(Response)

We are really thankful for highlighting the strengths of our study. We have carefully reviewed the limitations and we have revised the manuscript accordingly.

Point P2

The study’s weaknesses stem mainly from design limitations. The sample size is very small (N = 56) and single-center, which limits the generalizability of the findings. The cross-sectional design precludes assessing prognostic value over time.

(Response)

We deeply appreciate the reviewer’s suggestion. We fully agree with the important limitations and these limitations have been addressed in the limitation section of our manuscript. We sincerely appreciate reviewer’s valuable comments.

Line 282-288: “Third, because the number of participants included in this study was very small and the study was conducted at a single center, the external validity is limited and generalizability of the findings is restricted. In addition, due to the cross-sectional design of our study, it is not possible to clearly evaluate changes in G6PD activity before and after the development of bone metastasis. Further analysis in a large cohort of patients is needed to analyzed temporal changes in G6PD and the relationship with disease progression.”

Point P3

The authors lacked key confounding variables that could affect G6PD activity—such as infections, diabetes, hemolytic anemia, and chemotherapy—which they acknowledge; the lack of control for these factors limits the credibility of the conclusions. 

(Response)

We are really thankful for the reviewer’s insightful comment. We also consider the lack of inclusion of important covariates as a significant limitation of our study. Due to limitations in the available data, we were unable to include these variables in the current analysis. Future studies should take these variables into account to conduct more comprehensive and improved analyses. We also have added the limitation in the discussion section.

Line 270-277: “Despite its strengths, this study has several limitations. First, we were unable to consider multiple covariates that may influence G6PD activity due to insufficient data. According to previous studies, infection, diabetes mellitus, hemolytic anemia, TNM stages and chemotherapy status could vary G6PD activity [32-34]. The lack of consideration of covariates that could affect G6PD levels has limited the internal validity of this study. Multivariable logistic regression should be performed to adjust for these covariates in future studies. Also, further investigation involving subgroup analysis stratified by PSA levels and Gleason score which is key prognostic markers for prostate cancer is needed.”

Editor Point P4

Another limitation is that G6PD was measured in erythrocytes rather than serum or tumor tissue; as the authors correctly note, this makes the biological interpretation challenging since it is unclear whether RBC activity reflects tumor metabolism.

(Response)

We really appreciate for the reviewer’s comment about the biological mechanism. We also agree with that analyzing G6PD activity in tumor tissue would be more appropriate for biological interpretation. This point has been addressed in the limitation section of our manuscript. We sincerely appreciate reviewer’s valuable comment.

Line 277-282: “Second, the G6PD activity measured in this study was obtained from red blood cells (RBC). Nevertheless, there are biological limitations in explaining the relationship between cancer progression and G6PD activity measured in RBC, as opposed to G6PD activity within the cancer cells. Further research is needed to clarify the biological relationship between serum G6PD activity and tumor cell metabolism.”

Editor Point P5

Additionally, several authors have commercial ties to the production of the G6PD test, and a related patent has been filed—this represents a significant conflict of interest that could influence data interpretation.

(Response)

We are grateful for reviewer’s comment. We also consider potential conflicts of interest to be an important aspect. First, we have submitted a COI disclosure to Cancers, which included detailed information on conflicts of interest and authors’ signatures. Additionally, we have clarified that these conflicts did not influence the analysis.  

Line 336-341: “One author (Chang-Hwan Yeom) is currently involved in the outsourced production of a G6PD activity testing device and holds commercial rights related to its distribution and sales. This study was conducted independently, without any influence from potential conflicts of interest. Additionally, all participants included in this study were patients who received Vitamin C treatment. Also, the authors have filed a patent application related to the use of G6PD activity testing for the pre-diction of bone metastasis in prostate cancer.”

 Editor Point P6

Finally, the PPV values are low, which limits the clinical utility of the test as a standalone screening tool.

(Response)

We deeply appreciate for reviewer’s comment on the clinical utility. We also agree that the PPV is low (0.54). Accordingly, we have added that its limits the use of the test as an independent screening tool. We sincerely appreciate reviewer’s comment on this important limitation.

Line 212-214: “Furthermore, the positive predictive value is low, which limits its utility as independent screening tool for predicting bone metastasis.”

 Editor Point P7

In summary, the study presents an interesting and promising biomarker with solid biological rationale and consistent data, but its significance is constrained by small sample size, lack of clinical confounder control, the biomarker’s measurement method, and potential conflicts of interest. The results should be considered preliminary and require confirmation in larger, prospective studies.

(Response)

We deeply appreciate for reviewer’s comment. We have also added that the limitation should be addressed in a large cohort study. We are really grateful for reviewer’s comment.

Line 282-288: “Third, because the number of participants included in this study was very small and the study was conducted at a single center, the external validity is limited and generalizability of the findings is restricted. In addition, due to the cross-sectional design of our study, it is not possible to clearly evaluate changes in G6PD activity before and after the development of bone metastasis. Further analysis in a large cohort of patients is needed to analyzed temporal changes in G6PD and the relationship with disease progression.”

  Reviewer 3 Report

Comments and Suggestions for Authors

This article is interesting and addresses current and innovative topics in the field of prostate cancer management. However, some improvements could be made:

Comment 1: The first paragraph is redundant. It might be useful to keep only the abstract to reduce the repetition of concepts.

Comment 2: In the Introduction section, and especially in the Discussion, the relationship between G6PD activity measurement and PSA measurement should be described more clearly. In particular, it should be clarified what the limitations and advantages are of measuring one versus the other. Because while it is true that the diagnostic accuracy of PSA can be compromised by benign conditions, it is also true that G6PD activity may increase due to other unrecognized tumors.

Comment 3: Figure 1 should be adjusted to improve readability.

Author Response

Reviewer #3:

We deeply thank you for taking the time to read our manuscript and to provide feedback. Your comments were highly insightful and helped us to improve the quality of our study. We addressed the concerns point by point in the following pages.

 Point P1

The first paragraph is redundant. It might be useful to keep only the abstract to reduce the repetition of concepts.

(Response)

We are really grateful for reviewer’s comment. We fully agree that the abstract and the first paragraph contained overlapped content. We have revised the first paragraph by removing the redundant parts. We are thankful for reviewer’s valuable advice on improving the coherence of the manuscript.

Line 47-56: “In the United States, prostate cancer is the most common diagnosed type of cancer among men with approximately 300,000 new cases reported in 2024 [1] When prostate cancer metastasized to the bone, patients may experience various symptoms, including bone pain, along with skeletal-related events (SREs) such as pathological fractures and hypercalcemia [2,3]. In particular, pathologic fractures have been shown to negatively im-pact survival in patients with prostate cancer [4,5]. Also, early initiation of bone-targets therapy can help prevent SREs [6,7]. Thus, accurate prediction of bone metastasis in prostate cancer is essential for guiding treatment decisions, such as aggressive pain management, radiotherapy, and systemic therapies, and is also important for the prevention and management of related complications [8].”

 Point P2

Methods:In the Introduction section, and especially in the Discussion, the relationship between G6PD activity measurement and PSA measurement should be described more clearly. In particular, it should be clarified what the limitations and advantages are of measuring one versus the other. Because while it is true that the diagnostic accuracy of PSA can be compromised by benign conditions, it is also true that G6PD activity may increase due to other unrecognized tumors.

(Response)

We deeply appreciate the reviewer’s suggestion. We also agree that the advantages and limitations of both G6PD and PSA should be clearly described. We recognize that a major limitation of G6PD is its potential to increase in the presence of other undetected malignancies. We have incorporated this point into the discussion section. We are grateful for reviewer’s advice.

Line 196-214: “Several other biomarkers and clinical parameters have been investigated for their potential to predict bone metastasis in prostate cancer [19]. Bone-specific alkaline phosphatase (BALP) and pro-collagen type I N-terminal propeptide (PINP) are well-established bone formation markers that have been extensively studied in the context of predicting bone metastasis [9]. BALP has demonstrated a sensitivity and specificity of 0.75 and 0.93, respectively, while PINP has shown values of 0.78 and 0.68 [20,21]. Both markers exhibit-ed lower sensitivity compared to serum G6PD activity observed in the present study. On the other hand, prostate-specific antigen (PSA), a widely used biomarker in prostate cancer management, has demonstrated a sensitivity of 0.85 and a specificity of 0.84, both of which are higher than those of G6PD activity. However, its diagnostic accuracy can be compromised by benign condition such as benign prostatic hyperplasia, limiting its positive predictive value (PPV) in detecting bone metastasis [16]. In addition, Aihara et al. reported that PSA levels decrease in prostate cancers with a high Gleason score, which limits its ability to predict the progression of the disease[22]. G6PD activity has the advantage of predicting bone metastasis of prostate cancer through a simple blood test. However, a limitation remains in that G6PD levels may also increase in the presence of other undetected malignancies not related to prostate cancer. Furthermore, the positive predictive value is low, which limits its utility as independent screening tool for predicting bone metastasis.”

Point P3

Figure 1 should be adjusted to improve readability.

(Response)

We are really thankful for the reviewer’s insightful comment. We also agree with the comment that the readability of Figure 1  should be improved. First, we revised the legend to make it more specific and clearly. Second, we clarified the x-axis label. Finally, we increased the font size to enhance overall readability. We sincerely appreciate your helpful suggestions, which allowed us to improve the clarity of the figure.

Line 151-152: “

Round 2

Reviewer 1 Report

Comments and Suggestions for Authors

The author has successfully addressed all the reviewer’s comments and incorporated the suggested revisions. The manuscript now meets the required standards and is accepted for publication.